# Therapeutic Potential of Niche-Specific Mesenchymal Stromal Cells for Spinal Cord Injury Repair

**DOI:** 10.3390/cells10040901

**Published:** 2021-04-14

**Authors:** Susan L. Lindsay, Susan C. Barnett

**Affiliations:** Institute of Infection, Inflammation and Immunity, University of Glasgow, Sir Graeme Davies Building, 120 University Place, Glasgow G12 8TA1, UK; Susan.Barnett@Glasgow.ac.uk

**Keywords:** cellular niche, mesenchymal stromal cells, spinal cord injury

## Abstract

The use of mesenchymal stem/stromal cells (MSCs) for transplant-mediated repair represents an important and promising therapeutic strategy after spinal cord injury (SCI). The appeal of MSCs has been fuelled by their ease of isolation, immunosuppressive properties, and low immunogenicity, alongside the large variety of available tissue sources. However, despite reported similarities in vitro, MSCs sourced from distinct tissues may not have comparable biological properties in vivo. There is accumulating evidence that stemness, plasticity, immunogenicity, and adaptability of stem cells is largely controlled by tissue niche. The extrinsic impact of cellular niche for MSC repair potential is therefore important, not least because of its impact on ex vivo expansion for therapeutic purposes. It is likely certain niche-targeted MSCs are more suited for SCI transplant-mediated repair due to their intrinsic capabilities, such as inherent neurogenic properties. In addition, the various MSC anatomical locations means that differences in harvest and culture procedures can make cross-comparison of pre-clinical data difficult. Since a clinical grade MSC product is inextricably linked with its manufacture, it is imperative that cells can be made relatively easily using appropriate materials. We discuss these issues and highlight the importance of identifying the appropriate niche-specific MSC type for SCI repair.

## 1. Introduction

Spinal cord injury (SCI) results in devastating loss of sensory, motor, and autonomic function which leaves a sufferer incapacitated, bound to a wheelchair, and robbed of independence. SCI is therefore associated with substantial levels of suffering and a high socioeconomic burden. Clinical outcomes depend on the severity and location of the lesion but may include partial or complete loss of function below the level of injury. Trauma, the most common cause of SCI in the Western world, results in a complex injury pathology. The initial mechanical insult causes haemorrhage, shearing of axons, destruction of cells, and triggers a cascade of secondary injury mechanisms that lead to further cell death and demyelination [1]. Injury triggers an immune response, the formation of a glial scar, and, over the longer term, development of fluid filled cystic cavities [2,3]. As a result, the loss of function after SCI is generally permanent, and since there are currently no effective treatments, it represents a major unmet clinical need.

There are numerous strategies currently reported for the treatment of SCI; however, stem cell transplantation has gained the most interest [4,5]. The transplantation of autologous or allogeneic cells, including differentiated glia and various stem cells, have been extensively explored [6,7,8,9,10]. The aim of cellular transplantation is that it creates an environment favourable to repair, one which may offer neuroprotection, immune regulation, eventual axonal regeneration, neuronal circuit formation, and myelin regeneration [11,12,13]. One popular cellular candidate are mesenchymal stromal cells (MSCs). MSCs can be isolated from numerous tissue sources, are relatively easy to expand in vitro, and have unique beneficial immunological properties [14]. Furthermore, the MSC secretome, which has a paracrine effect on the local environment after injury, have made them well suited to SCI repair (Figure 1). MSCs were first identified in 1968 by Friedenstein and colleagues as a population of adherent cells present in the bone marrow (BM) which exhibited fibroblast-like morphology [15,16]. Today, numerous other tissue sources have been identified, such as adipose, dental pulp, periodontal ligament, tendon, skin, muscle, and newer tissues, such as the olfactory mucosa and lung [17,18]. There are likely many other sources within the human body; however, practical limitations rule them out as potential therapeutic sources [18,19]. MSCs are defined by their capability to self-renew, ability to differentiate into three specific cell lineages in vitro (osteoblasts, adipocytes, and chondrocytes; although in vivo they can make other stromal cell types, including endothelial cells) and by their expression of a subset of cell surface proteins; a specific MSC marker has yet to be identified [20]. The International Society for Cellular Therapy has stated that MSCs should express CD105, CD73, and CD90 and lack expression of CD45, CD34, CD14, CD11b, CD79a, or CD19, and human leukocyte antigen-DR (HLA-DR) surface molecules.

There is a wealth of literature describing the promising effects of MSC therapy after SCI. In general, they have both an anti-inflammatory and pro-regenerative capacity in human and animal models, which has been covered extensively in other recent reviews [21,22]. Importantly, their use in phase I/II clinical SCI trials have confirmed their safety [21,23,24,25]. However, whilst many clinical trials report promising efficacy for sensorimotor function, the risk of bias is high since few studies have included control groups, and the study subject number is low [24]. Many aspects concerning MSC therapy require clearer definition if they are to be fully translated to the clinic. For example, MSCs derived from specific cellular niches may have potential limitations due to niche-specific inherent properties. Understanding the MSC role both within their resident niche and their fate after SCI will improve their use as a therapy. Tissue-specific stem cells support the tissue type from which they originate, meaning that specific MSC types might be more suited for the treatment of SCI than others. In addition, the ease of accessibility of the anatomical location of specific MSC types may be more practical for clinical translation. Crucially, the generation of MSCs will need to follow strict good manufacturing practice (GMP) guidelines to ensure the safety and quality of the end-product before use in clinical trials. Therefore, culture reagents used in their isolation and maintenance must meet these guidelines, and the impact of changing protocols to one more GMP compliant should be considered. Here, we focus on the most commonly trialled MSCs for the treatment of human SCI around the world: those derived from autologous bone marrow (BM-MSCs), umbilical-cord-derived mesenchymal stem cells (UC-MSCs), and adipose-derived mesenchymal stem cells (AD-MSCs) [25]. In addition, MSCs isolated from human olfactory mucosa will be discussed as cells derived from this tissue are also one of the most frequently trialled for CNS repair [25].

## 2. Bone Marrow Mesenchymal Stromal Cells (BM-MSCs)

Bone-marrow-derived MSCs (BM-MSCs) are the most extensively studied cells both in vitro and in vivo. Currently there are 20 SCI trials registered on the ClinicalTrials.gov database using BM-MSCs, which makes them the most clinically applied of all the MSC types [26]. The BM niche integrates endocrine, autocrine, and paracrine signalling in order to sustain the stem cell pool [27]. The niche is a complex mix of heterogeneous populations of stromal cells, which interact with hematopoietic stem progenitor cells (HSPCs). BM-MSCs secrete soluble factors which control HSPC maintenance and fate, such as stem cell factor (SCF), stromal cell-derived factor (SDF-1 or CXCL12), bone morphogenetic protein 4 (BMP-4), transforming growth factor (TGF)-β, leukaemia inhibitory factor (LIF), and other cytokines that influence mature hematopoietic progenitors, such as, granulocyte macrophage colony-stimulating factor (GM-CSF) and granulocyte colony-stimulating factor (G-CSF) [28]. BM-MSCs also produce several interleukins, for example, IL-1, IL-6, IL-7, IL-8, 1L-11, IL-12, IL-14, and IL-15 [29]. In vivo, HSPCs are found in distinct locations within the BM—the majority within sinusoidal blood vessels, which is considered a perivascular niche. However, <20% are found close to the endosteum [30]. Since HSPCs localise within the BM according to their stage of differentiation, it is likely the features of BM-MSCs that are vital to maintenance will also vary dependent upon their location.

Interestingly, a nestin-expressing MSC population (Nestin+ MSC) that is closely associated with HSPCs in perivascular regions has been identified [31]. Nestin+ MSCs are typically associated with adrenergic nerve fibres of the sympathetic nervous system that regulate HSPC mobilisation [32]. Nestin+ MSCs show multilineage differentiation and self-renewal ability and express higher levels of HSC maintenance factor transcripts, including CXCL12, SCF, angiopoietin-1 (Ang-1), IL-7, vascular cell adhesion molecule 1 (VCAM1), and osteopontin (OPN), compared with other BM stromal cell types. Whether Nestin+ MSCs are better suited for SCI repair over BM-MSCs is yet to be established; however, they improve cardiac function in an acute myocardial infarction model via the CXCL12/CXCR4 chemokine pathway [33]. Since Nestin+ MSCs are present in low frequency within the endosteum, harvesting them in numbers large enough for clinical translation is more difficult, impeded further by the requirement of more restrictive isolation protocols. In addition, it is still unclear whether the different niche populations described in mice are preserved in humans.

BM-MSCs express low levels of human leukocyte antigen class II (HLA-II) and co-stimulatory molecules CD40, B7, CD80, and CD86, making them well known for their immunomodulatory behaviour [34]. That being said, it is the immunological conditions of the local microenvironment that regulate this [29]. In particular, interferon (IFN)-γ and tumour necrosis factor (TNF)-α create a microenvironment leading to the induction of immunosuppressive BM-MSCs. Since inflammation is a key secondary pathological mechanism following SCI, it is not surprising that BM-MSCs have shown promising anti-inflammatory effects in animal models of SCI [35,36]. BM-MSCs strongly respond to inflammatory or chemotactic stimuli released from injured tissues, including chemokines and various growth factors, such as vascular endothelial growth factor (VEGF), hepatocyte growth factor (HGF), and in particular the CXCL12/CXCR4 pathway [26,27]. BM-MSCs can therefore migrate to the lesion site following SCI and enhance antiapoptotic effects. Consistently, the CXCL12/CXCR4 pathway enhances BM-MSC migration toward injured tissues and promotes recovery after SCI [29,30]. Indeed, impaired expression of CXCR4 leads to reduced homing and BM-MSC engraftment [28]. Hypoxia is also a critical element of the BM niche, not only for maintaining ‘stemness’, but also for inducing glycolytic pathways and inducing cellular responses that maintain the niche [37]. This hypoxic niche native to BM-MSCs is well suited to SCI repair since after injury hypoxia in the lesion causes progressive and irreversible damage to neurons. When in a hypoxic environment, Toll-like receptor ligands and trauma site inflammation factors induce MSCs to secrete factors that stimulate tissue regeneration, including epidermal growth factor (EGF), insulin-like growth factor (IGF), and fibroblast growth factor (FGF), which may significantly upregulate and promote angiogenesis and inhibit apoptosis [38].

### 2.1. Anatomical Origin and Culture

The iliac crest has become the standard site of BM harvest, which provides a modestly high concentration of nucleated cells [39,40] (Figure 2a). BM aspirate can be taken from the posterior or anterior iliac crest, although it is considered that the posterior provides a greater number of cells, likely associated with the required aspiration technique [40]. BM-MSCs can also be isolated from other bone sources, such as humerus, femur, tibia, vertebral body, or calcaneus [40]. Excitingly, BM has also been isolated from the vertebral body of patients undergoing posterior lumbar arthrodesis [41] and spinal fusion [42]—surgeries that newly injured SCI patients may undergo. It was reported that a similar number of cells were isolated compared with matched controls taken from the iliac crest [41]. This would make a promising alternative for SCI patients since it avoids the need for a second surgery. However, in other studies it has been reported that cell populations from different bones can vary in their concentration, prevalence, and biological potential, meaning that culture expanded BM-MSCs from different bone sources may be highly variable [39].

Within the human BM, mesenchymal progenitors only account for approximately 0.001% to 0.01% of all mononuclear cells. Therefore, their ex vivo expansion is necessary to reach the numbers of cells required clinically [43]. Adherent BM-MSCs outgrow any fully differentiated and non-proliferating cells present in the stroma, which may also adhere to tissue culture plates [44]. This is important since currently known BM-MSC markers are also expressed by other cell types found in the BM, be they of hematopoietic or endothelial lineage [45]. BM aspirate can produce 10–15 × 10^6^ BM-MSCs from 25 mL when cultured for 3 weeks, and this number of cells when packed into a cell slurry can create a volume of about 0.4−0.5 mL [14]. Since the majority of clinical trials using BM-MSC to date, have involved the administration of cells via an injection directly into the lesion site and/or proximal and distal to the spinal cord lesion [25], having a large enough volume of cells is of vital importance, especially given the variability of lesion size from patient to patient. A typical cavity in rat SCI models is 1–3 mm in length into which, generally, 1–5 × 10^6^ cells is required to fill it. The human equivalent will therefore require scaling up. However, the isolation, expansion, validation, and production of BM-MSCs for clinical use has already been established, paving an easier transition to the clinic than other MSC types [46].

### 2.2. Parameters Related to Donor of MSCs

There are potential limitations of using autologous BM-MSCs derived from SCI patients, as reports suggest that the isolation efficiency is related to the age of the donor tissue [47]. BM-MSCs obtained from younger donors have a higher proliferation rate and are less susceptible to oxidative damage, whereas BM-MSCs isolated from elderly donors have decreased biological activity, including reduced differentiation and regenerative potential [10,11]. A challenging issue is how to expand autologous BM-MSCs from older SCI patients to efficient numbers of therapeutic cells. In addition, the loss of mechanical loading following SCI is thought to be a crucial stimulus for bone resorption [48]. In the 12–25 weeks after SCI, trabecular bone volume has been reported to decrease by 30% [44]. Moreover, there is a significant change in the composition of iliac crest tissue in individuals with complete paralysis compared with non-SCI donors [49]. Such changes in the BM microenvironment may have an impact on cells resident within it. In fact, SCI mice acquire BM failure, leading to intrinsic long-term functional impairment of HSPCs [50]. Therefore, the acquired effects on the BM after injury may preclude the use of BM-MSCs from SCI donors as a transplantation source [50].

## 3. Adipose-Derived Mesenchymal Stromal Cells

Human adipose-derived stem cells (AD-MSCs) have also gained a great amount of attention, since they can be easily accessed via subcutaneous lipoaspiration, which is less invasive compared to the harvest of BM-MSCs. In the case of allogeneic AD-MSCs, there are few ethical issues associated with the use of autologous fat or routine fat waste generated during liposuction surgeries. Adipose tissue originates in the mesoderm and comprises different depots, some of which contribute to the homeostasis of various tissues, such as BM, skin, and blood vessels [51]. AD-MSCs are located within the stromal vascular fraction (SVF), a niche that contributes to the local regulation of angiogenesis and vessel remodelling [52]. In normal conditions, the niche supports angiogenesis through VEGF and platelet-derived growth factor (PDGF) signals [52]. AD-MSCs enable tissue repair but also regulate inflammation and can modulate both innate and adaptive immunity. AD-MSCs represent approximately 50% of the cells within the SVF, which is almost 500-fold greater than BM-MSC expression within their relative niche [53]. This means AD-MSCs could be used clinically without expansion if a large enough volume of lipoaspirate was harvested. The rest of the SVF fraction is a heterogeneous mix containing pre-adipocytes, endothelial cells, erythrocytes, fibroblasts, pericytes, T cells, and macrophages—all potential contaminating cell populations [54]. However, there is evidence that the cellular components of SVF may act synergistically with AD-MSCs and be better transplanted in combination than alone [55]. Indeed, the AD-MSC niche is a key component in cell fate determination and the intrinsic adipocyte capacity [56]. Numerous factors are involved, including the interaction between AD-MSCs, or their interaction with other types of cells, the extracellular matrix (ECM), growth factors, oxygen tension, cytokines, pH, and ion concentrations [56].

There are two main regions of white adipose tissue: subcutaneous adipose tissue (SAT) and visceral adipose tissue (VAT). Different regions have distinct functions in adipose biology, making it likely that derived cells have differing intrinsic functions. Not surprisingly, there are differences in the genetic profiles, secretome, and stem cell markers of AD-MSC isolated from different adipose regions [57]. These variations are due to the different developmental origin and/or the ECM and microenvironment that they exist in [58,59]. In the normal state, SAT-derived adipocytes are relatively small in size and secrete leptin; in contrast, VAT adipocytes are larger and secrete adiponectin, fibronectin 1, and laminin [60]. Therefore, different sources of adipose tissue have a different microenvironment. In fact, the composition of the surrounding ECM of adipose tissue changes during development and is also different between the different depots [61]. The ECM of adipose tissue is a complex mix of proteoglycans, polysaccharides, different collagen types, elastins, fibronectins, and laminins—each with a specific structural function and binding ability to cellular adhesion receptors [61,62]. It therefore serves as a reservoir for various growth factors which can then modulate AD-MSC function, making them highly sensitive to their surrounding ECM [61]. The main function of the niche is to protect the adipocyte stem cell pool from overactivity, proliferation, or differentiation, which can be harmful. It is known that adipogenesis and the production of adipose-derived regulatory factors are compromised in several conditions, such as chronic vascular dementia, neurodegeneration, cardiovascular disease, and Alzheimer’s disease [63,64].

AD-MSCs have been found on the outer adventitia of blood vessels [65,66], and since they express CD146, 3G5, and other markers that typically define pericytes, there have been suggestions that they are pericytes [67,68]. Interestingly, they secrete higher levels of CCL2, CXCL12, PDGF-β, and VEGF-A [69], along with brain-derived neurotrophic factor (BDNF), VEGF, and HGF compared to BM-MSCs [70]. In addition, they tolerate hypoxic conditions better than BM-MSCs, which may make them more suited to SCI repair [69]. It has been reported that AD-MSCs are HLA-DR negative and retain a lower HLA-DR expression under inflammatory conditions compared with BM-MSCs, important for their future clinical use in allographs. In addition, AD-MSCs immunomodulatory effects exceed those of BM-MSCs due to their higher level of cytokine secretion [71,72]. However, there is evidence that their immunosuppressive profile may not be constitutive and depends on the pathophysiologic microenvironment they are exposed to [51]. Tissue damage or disease can cause AD-MSCs to undergo hyper-proliferation, leading to dysfunctional cells [73,74]. This suggests that the fine control the niche has over AD-MSCs is of vital importance. In the context of SCI, AD-MSCs may not be suited to the inflammatory environment immediately after injury and may be better considered for transplantation at a chronic time point.

### 3.1. Anatomical Location and Culture

There are two clinically relevant depots: the SAT and the omental region of VAT [57] (Figure 2b). Before the fat can be aspirated, the area is infiltrated using lidocaine, epinephrine, sodium bicarbonate, and saline (to reduce the risk of blood loss). Subsequently, the separated fat tissue is aspirated using a cannula. However, local anaesthetic reagents negatively impact both the viability and quantity of AD-MSC harvested [75,76,77,78]. In addition, cannula diameter can influence surgical position during liposuction; a smaller diameter may be preferentially oriented toward superficial fat layers, whereas a larger cannula may be placed toward deeper fat layers [79]. The specific orientation of the cannula may then affect the number of AD-MSCs collected because SVF is more concentrated in superficial fat layers [65]. Nevertheless, a single liposuction procedure produces litres of fat, and only one millilitre is enough to produce 250,000 AD-MSCs by a single passage, a major advantage to their use [80]. Their isolation from the stromal fraction can be carried out using either enzymatic or nonenzymatic dissociation using a manual or automated system. The most widely used in vitro method consists of lipoaspirate washing, followed by enzymatic digestion with collagenase, centrifugation, and red blood cell lysis [54]. However, xeno-free enzymatic products have replaced research grade products effectively without any negative effect in the yield or function of AD-MSCs [54]. When grown in vitro, AD-MSCs have a higher proliferative capability compared to BM-MSCs, with a faster population doubling time (range 40–120 h), but this is dependent upon a variety of factors, such as donor age, type/location of the adipose tissue, method of collection, culture conditions, cell density, and medium composition used [54,81,82]. In addition, AD-MSCs are genetically and morphologically stable in long-term culture [83], display a lower senescence ratio [83,84], and retain differentiation potential for a longer period in culture when compared to BM-MSCs [83]. However, it has been reported that AD-MSCs exhibit a higher DNA damage response than BM-MSCs due to their higher proliferative phenotype [73].

### 3.2. Parameters Related to Donor of MSCs

Relevant therapeutic numbers of AD-MSCs may be difficult to obtain from patients with low levels of subcutaneous fat. In addition, people with obesity have significantly impaired proliferation and differentiation potential in isolated AD-MSCs [16]. Biopsies from obese human subjects have revealed differing expression of extracellular matrix proteins in the VAT, compared to those specifically expressed in SAT [85]. Since individuals with chronic SCI are susceptible to central and visceral obesity, the impact this may have on isolated cells must be considered [86]. In particular, abdominal obesity within VAT increases following SCI, which is a clinically relevant region for AD-MSC harvest. Obesity stimulates changes in various types of leukocytes that reside in adipose tissue, elevating the expression levels of inflammatory cytokines and adipokines [87]. Nevertheless, obese patients also show altered haematopoiesis within the BM, which is characterised by an accumulation of pro-inflammatory immune cells [71]. The impact this may have on harvested cells should be fully considered prior to the use of autologous AD-MSCs or BM-MSCs for clinical use.

## 4. Umbilical-Cord-Derived Mesenchymal Stromal Cells (UC-MSCs)

An alternative source of allogeneic MSCs can be derived from young ‘adult’ birth-associated tissues, such as umbilical cord (UC) [14]. They are the second most used MSC type in SCI trials after BM-MSCs, with 15 currently listed on the ClinicalTrials.gov database. Fibroblast-like cells were originally identified in the mucous connective tissue of human UC, known as Wharton’s jelly (WJ) [88,89]. Since then, MSC-like cells derived from human UC tissue or blood have been shown to have similar surface phenotype and plastic adherence as MSCs derived from other sources [90]. UC-MSCs are multipotent, self-renewing, and can differentiate into ectoderm- and mesoderm-derived cells [91]. The role of UC-MSCs within the UC are not clear. Although they are thought to circulate in the blood of preterm foetuses, it is not known where UC-MSCs home to at the end of gestation [92]. Like other MSC types, they modulate the immune response. In vitro they reduce inflammation via their secretion of IL-10 and IL-4 and have paracrine effects via the secretion of keratinocyte growth factor (KGF), HGF, EGF, and other cytokines [91]. In addition, their anti-inflammatory properties in an in vivo model of SCI were attributed to a reduction in IL-6, IL-7, and TNF-α at the injured site [91]. In terms of their therapeutic action, after SCI they also have the potential to differentiate and replace damaged tissue and cells. Relevantly, UC-MSCs can be induced to differentiate in vitro into neural cells [7,16,17,18], which give ideal candidature for treating patients with neurodegenerative diseases or CNS injuries. Furthermore, their very low surface antigen expression lowers the risk of transplantation rejection, facilitating their use in allografts. UC-MSCs express pluripotency gene markers Oct-4, nanog, and Sox-2 at lower levels than found in embryonic stem cells (ESC) but higher than those found in BM-MSCs, perhaps indicating they are a subset of primitive stem cells [93,94].

### 4.1. Anatomical Location and Cell Culture

The UC contains two umbilical arteries and one umbilical vein, both embedded within WJ, which is then covered by amniotic epithelium. WJ is rich in proteoglycans and hyaluronic acid (HA), whose function is to insulate and protect umbilical vessels and ensure a constant blood flow between foetus and placenta [95]. For therapeutic purposes, MSCs have been isolated from four different regions: whole umbilical cord tissue, WJ, UC blood or from the UC tissue vessels [96] (Figure 2c). Differing functional characteristics have been reported for the various types; however, whether this a result of the isolation and expansion methods used or due to the different cord tissue compartments source is still unknown. UC-blood-derived MSCs may have a limited potential due to their relatively low expression compared to BM (approximately 0.000001–0.001% compared to 0.001–0.01% within BM) [95]. In contrast, the frequency of MSCs in other UC-tissues is thought to be much higher. WJ-MSCs can be separated with minimal contamination from the constituents of the UC, and harvested cells have faster expansion rates, doubling times, and purity than UC-blood-derived MSCs [97]. Still, once harvested UC-MSCs have greater ex vivo expansion capabilities and proliferate faster than either BM-MSCs or AD-MSCs, retaining stemness and multilineage differentiation capacity after multiple passages [84,98]. Comparison of MSCs from BM, UC, and UC blood found all expressed similar markers, yet only UC-MSCs and UC blood MSCs did not express HLA-DR associated with transplant rejection, making them more similar to AD-MSCs [99]. Although, UC-MSCs express nestin early after isolation, their expression declines with passage [100,101]. This reduction in nestin expression correlated with reduced anti-inflammatory capacity later in passage [101]. In addition, cell surface marker expression is thought to change in passage and may indicate epigenetic phenomena associated with cell culture [102].

There are two main methods to obtain UC-MSCs: one which employs explanting UC tissue and the other which uses enzymatic digestion. In the explant method, UC, or its compartments, are minced into small fragments 1–2 mm^3^ and seeded onto tissue culture-treated dishes. Fibroblast-like adherent cells naturally migrate from the explanted tissue and reach 80–90% confluency in about 2–4 weeks. The adherent cells are detached using a trypsin solution and filtered to remove any tissue fragments. The disadvantage of this method is that the fragments often float in the medium, resulting in poor unreproducible cell recovery. Alternatively, the tissue can be digested using proteolytic enzymes, such as collagenase and hyaluronidase [103]. This method allows for a more rapid acquisition of cord tissue cells, although the cell population generated will be mixed, meaning additional steps must be taken to enrich for UC-MSCs [103]. In direct comparisons, explant cultures provided a higher number of isolated cells, with higher proliferative rates and purer cell populations [104]. There is also the risk that enzymatic digestion damages the cells during the process. Like other MSC types, UC-MSCs are traditionally expanded using media containing serum, which due to the presence of xenogeneic materials is not compliant with future GMP manufacture. Clinically, the use of human autologous serum and/or platelet lysate can be used instead of serum, and UM-MSCs grown in UltraGRO^TM^, a xenofree alternative to FBS, have better anti-inflammatory performance, correlating with retained nestin expression [101]. In theory, these cells may be stored to provide stem cells for therapeutic use decades later, though the phenotype would need to be tightly defined and reproduced post thaw to meet GMP compliance.

### 4.2. Parameters Related to Donor of MSCs

The collection of MSC-like cells from tissues that are discarded at birth is easier and less expensive than collecting MSCs from BM aspirate. In contrast to autologous tissue, the acquisition of birth-associated tissues does not pose a risk of complications for the donor, as could be the case for autologous tissue. While allogeneic-derived UC-MSCs could be considered advantageous over autologous MSC sources, in vivo studies have reported that allogeneic MSCs are not immune privileged to the same extent and cause an immune response, despite the immunosuppressive properties and low immunogenicity documented in vivo and in vitro [105]. Recent studies suggest that MSCs may not be ‘immune privileged’ and are no longer considered immunologically silent in vivo [17,18]. Therefore, the use of allogeneic MSCs has limitations, considering the risk of inducing an immunological response [19]. Therefore, the decision to use autologous or allogeneic MSCs is a fundamental clinical decision, though the risk of an immunological rejection must be considered [14]. UC blood, UC-MSCs, or UC fragments have the potential to be banked and thus are available for use as an ‘off-the-shelf’ therapy; however, confirmation of the baby donor health should be verified post-tissue acquisition, and thus genomic or chromosomal tests need to be performed [106]. In the case of UC banking, many banks continue to monitor the baby’s health after birth. Nonetheless, an important advantage in the use of UC-MSCs is the avoidance of potential problems related to donor age, which is well known for BM-MSCs [106].

## 5. Olfactory-Mucosa-Derived Mesenchymal Stromal Cells (OM-MSCs)

Cells derived from the primary human olfactory system have also been popular for transplant-based therapies for SCI due to the unique reparative properties of this tissue [25]. Clinical studies have used mixed suspensions of olfactory cells [107,108] or in some cases whole, undissociated pieces of mucosal tissue [109,110,111] as a source of cells. Recently, a population of MSCs were isolated and characterised within the mucosa, termed olfactory-mucosa-derived mesenchymal stromal cells (OM-MSCs), which possess many therapeutic benefits for SCI repair [112]. OM-MSCs were first identified as a population of multi-potent cells with MSC characteristics in the rodent [113]; however, the human equivalent has since been extensively studied in vitro [114,115,116]. The peripheral olfactory system in which OM-MSCs reside comprise olfactory receptor neurons (ORNs) responsible for detecting odour molecules. The ORNs transmit the information via the glomerular layer in the olfactory bulb to second-order neurons in the brain, allowing the perception of the sense of smell [117]. The olfactory mucosa is therefore a neurogenic niche vulnerable to physical and chemical injury (e.g., inhalation of a noxious substance). Nonetheless, it can undergo continuous cell replacement after injury, a process supported by epithelial-resident basal stem cells, a specialised glial cell type termed olfactory unsheathing cells (OECs), and the surrounding niche. Olfactory stem cells are unique in that they regenerate the entire damaged epithelium, including new ORNs, meaning that this tissue undergoes continued neurogenesis throughout life [118]. Because human ORNs are the only accessible neuronal cells, the OM has been considered as a ‘window to the brain’ [119] since biopsy of these cells could help early diagnosis of neurodegenerative conditions, such as schizophrenia, Alzheimer’s disease, multiple sclerosis, and Parkinson’s disease [120]. Olfactory stem cells receive and respond to various feedback signals from their immediate environment and react to the changing needs of the tissue. OM-MSCs reside within the underlying lamina propria and may have a paracrine role over neighbouring niche cells, which include their interaction with the above neuroepithelium. Molecular signalling between neuroepithelium and lamina propria influences the olfactory pathway development [121]. Interestingly, OM-MSC transplantation into a mouse lesioned hippocampus showed their ability to differentiate into neurons and restore long-term memory [122]. It has therefore been postulated that OM-MSCs can cross the basement membrane to differentiate into neurons and replenish the olfactory epithelium after extensive damage [116]. This is supported by their higher levels of transcripts normally expressed in neural cells when compared to BM-MSCs [116].

Even so, it is likely their paracrine effect on the local injury environment makes them suited to SCI repair [112]. OM-MSCs have immunosuppressive effects on microglia and secrete less of the pro-inflammatory cytokines, IL-6, IL-8, and CCL2 than BM-MSCs [115]. They suppress the cytotoxic function of CD8+ lymphocytes and natural killer cells, illustrating similar immunomodulatory function as other MSC types [123,124]. A role unique to OM-MSCs is their ability to promote oligodendrocyte differentiation and CNS myelination in vitro—features not associated with BM-MSCs [112,114]. Interestingly, the pro-myelinating effect of OM-MSCs was attributed to their secretion of greater levels of CXCL12 compared to BM-MSCs [115]. Furthermore, in vivo OM-MSCs promote Schwann cell remyelination and improve locomotion in an animal model of SCI [125]. The regeneration of ORNs that occurs continually within the olfactory mucosa may therefore be supported by OM-MSCs, which are suited to a neurogenic environment. Although OM-MSCs have a similar CD marker expression and are 68% homologous to BM-MSCs, they express 100% nestin immunoreactivity [17,126]. As stated, Nestin+ MSCs found within the BM also secrete high levels of CXCL12 and have been reported to be derived from the neural crest. This suggests that OM-MSCs are derived from a similar source, further supported by the fact that in reporter mice, OECs and other OM cell types originate from the neural crest [127,128,129]. Although it cannot be excluded that other MSC types that constitutively express higher levels of CXCL12 also promote CNS myelination, OM-MSCs are widely distributed throughout the highly accessible olfactory mucosa [130], which makes them a novel candidate for SCI repair.

### 5.1. Anatomical Location and Cell Culture

The human olfactory mucosa lines the dorsoposterior and superior turbinates and nasal septum and can be easily accessible through the nasal cavity [131] (Figure 2d). Biopsies taken from superior turbinates have been shown to be safe without any detrimental effect on a patient’ sense of smell [132,133]. The OM only covers approximately 3% of the total nasal surface (approximately 500 mm^2^), compared to approximately 50% for F344 rats, likely due to humans less developed olfaction [134]. This may make obtaining pure OM biopsies difficult; however, the extent to which this is an impediment to the culture of OM-MSCs is unclear. It is likely that OM-MSCs are dispersed throughout the entire lamina-propria of the respiratory epithelium, since lung-resident MSCs have recently been identified [135]. OM-MSCs reside in the highly cellular and heterogenous lamina propria, meaning their standard method of isolation requires collagenase digestion, followed by the plating onto collagen coated tissue culture flasks. Cells grown via this method or similar have been used in clinical trials, and the un-purified cell populations have been termed ‘OECs’ or ‘olfactory cells’, yet they are likely a heterogenous mixture containing OM-MSCs. Indeed, the proliferation rate of OM-MSCs is almost 8 times that of BM-MSCs and 4 times that of OM fibroblasts, so once culture expanded, they would outgrow any potential contaminating cell types [114]. Still, un-purified cellular mixes are not suitable for future clinical translation since autographs of olfactory cells have led to respiratory spinal masses in SCI patients, highlighting the importance of cellular purification [136,137]. OM-MSCs can be purified and enriched using specific isolation kits available based on the cell surface marker CD271 [138]. CD271 selectively isolates BM-MSCs with a higher clonogenicity, lower hematopoietic contamination, higher paracrine secretion of cytokines, and significantly pronounced lymphohematopoietic engraftment-promoting properties [53]. Within the OM niche, the only other cell type to express CD271 is OECs; however, human OM-derived OECs are notoriously difficult to expand in culture, and their growth conditions are yet to be fully determined [139]. The current method of OM-MSCs isolation is not currently compliant with future GMP manufacture. Still, with the development of GMP compliant procedures already proven for other tissue-derived MSCs, the transition to a method more suitable may be quick to establish [140]. In fact, early investigation using GMP compliant methods has already begun in collaboration with the Scottish National Blood Transfusion Service, UK, and revealed no effect on proliferation or cellular phenotype of OM-MSCs (Barnett lab, unpublished observations). The full characterisation of GMP compliant OM-MSCs and whether they retain their pro-repair phenotype post manufacture remains to be fully established.

### 5.2. Parameters Related to Donor of MSC

Disruption of the cells within the OE has been associated with age [141,142] and environmental insults; for example, viruses can cause cell replacement with non-neuronal respiratory epithelium [131]. This has potential impact on the harvest of OECs or other epithelial-derived stem cells, but importantly, the lamina-propria in which the OM-MSCs reside appears unaffected. Moreover, both in vitro and in vivo evidence of OM-MSCs pro-myelinating and pro-repair effects have come from OM tissue harvested from elderly patient donors undergoing routine polypectomy removal [112,114,115]. Even though OM-MSCs have high mitotic activity, they maintain self-renewal ability for long periods by conserving telomeric activity and inhibiting apoptotic activity, characteristics that are not affected by donor age [143]. This would suggest that OM-MSCs are not impacted to the same degree as other MSC populations by patient donor age, making them a suitable alternative for transplant-mediated repair.

## 6. Conclusions

In this review, we discussed the potential and limitations of various MSC types that have been documented in SCI clinical trials across the world (Figure 3 and Table 1). An ideal MSC candidate for transplant-mediated repair would be one that is able to modulate the inflammatory environment, promote myelination, have neurogenic properties, and preferably reside in a niche that is unaffected by the injury itself. In addition, cells that are easily harvested and grown quickly to clinically relevant numbers offer clear advantages. A targeted approach to the isolation of niche-specific MSCs more intrinsically suited for the treatment of SCI may offer better therapeutic benefits.

Different niche-derived MSCs may offer differing therapeutic benefits. BM-, AD-, and UC-derived MSCs have all reached clinical trial for the treatment of SCI across the world. Although olfactory tissue and cells have also reached clinical trial, purified OM-MSCs have yet to be tested. Consideration should be given to the impact of the procedure on the patient donor and whether enough cells will be generated quickly post tissue acquisition. Nestin expression on MSCs correlates with enhanced CXCL12 expression, which promotes myelination. This may be an important marker for isolating MSCs with repair characteristics better suited for SCI repair. Within the BM there is a small population of Nestin+ MSCs; however, these may be difficult to isolate. UC-MSCs express high levels of nestin soon after isolation, although this is lost after passage. AD-MSCs have not been found to express nestin but can upregulate it after neural differentiation in vitro. OM-MSCs constitutively express nestin which is not lost during passage. Key: +++ high, ++ moderate, + low.

## Figures and Tables

**Figure 1 cells-10-00901-f001:**
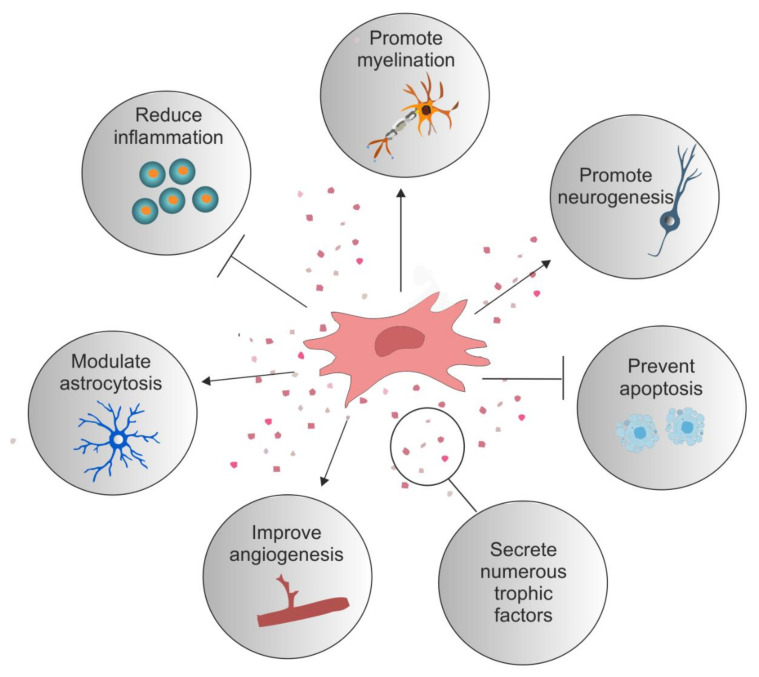
Reparative role of mesenchymal stromal cells. Examples of the reparative potential of mesenchymal stromal cells (MSCs) for spinal cord injury (SCI). MSCs secrete numerous trophic factors and anti-inflammatory molecules that change the injury milieu to pro-regenerative. These secreted factors have an anti-inflammatory effect on numerous immune cells such as T cells, B cells, macrophages, and microglia. They reduce astrocytosis and promote axonal growth and neuroprotection. They stimulate angiogenesis and offer protection against apoptotic cell death. Certain MSC types promote the differentiation of oligodendrocytes and myelination.

**Figure 2 cells-10-00901-f002:**
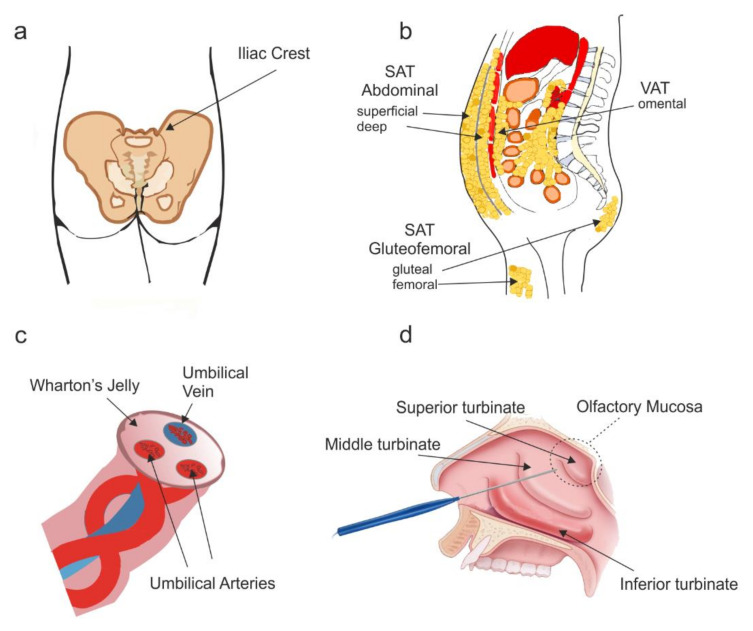
Anatomical location of routinely used MSC types for the treatment of SCI. Schematic diagrams detailing the anatomical locations of the most common MSC types used in clinical trials for the treatment of SCI. (**a**) Bone marrow (BM)-MSCs are routinely harvested from bone marrow aspirated from the iliac crest. (**b**) Adipose-derived (AD)-MSCs are generated from lipoaspirate collected from subcutaneous adipose tissue (SAT), or the omental region of visceral adipose tissue (VAT). (**c**) Umbilical cord (UC)-MSCs can be isolated from whole umbilical cord tissue, Wharton’s jelly, or the umbilical cord vein or arteries. (**d**) Olfactory mucosa (OM)-MSCs can be isolated from the olfactory mucosa which lines the dorsoposterior and superior turbinates and is accessible via the nasal cavity.

**Figure 3 cells-10-00901-f003:**
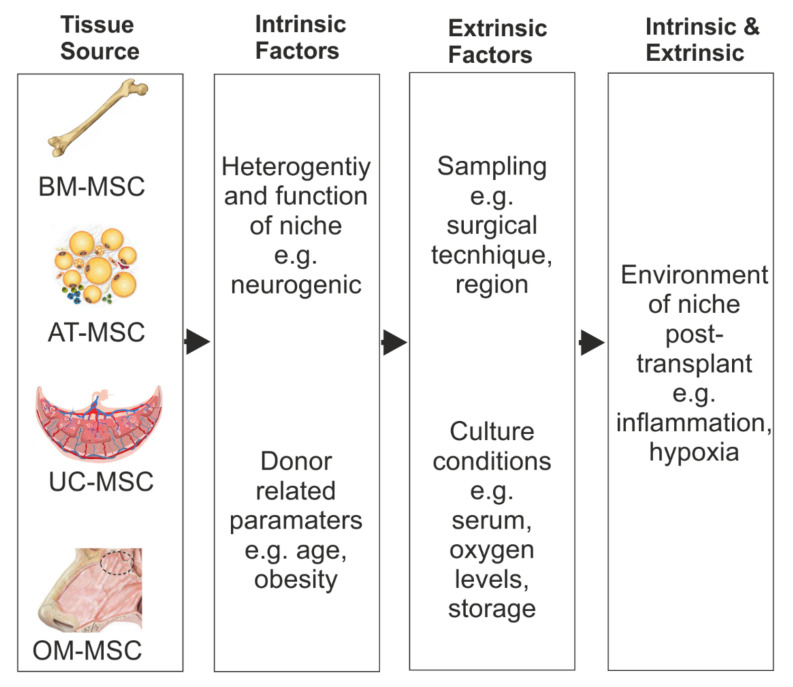
Factors which can affect MSC repair benefits. Schematic representation of the potential MSC heterogeneity because of their different tissue source and the intrinsic and extrinsic factors that could influence their repair benefits after SCI.

**Table 1 cells-10-00901-t001:** Comparison of key characteristics of MSCs harvested from different sources.

	BM-MSC	AD-MSC	UC-MSC	OM-MSC
Niche	Haematopoietic	Angiogenic	Haematopoietic	Neurogenic
Tissue Availability	++	+++	+++	+++
Use in SCI clinical trials	+++	++	+++	+
Procedure	Invasive	Minimally Invasive	Not Invasive	Minimally Invasive
Proliferative Capacity	+	++	+++	+++
Cell Yield	+	+++	++	+++
Autologous use	+++	+++	+	+++
Allogenic use	+++	+++	+++	+
Nestin expression	++	+	++	+++

## Data Availability

Not applicable.

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
