# Peer review of "Therapeutic Potential of Niche-Specific Mesenchymal Stromal Cells for Spinal Cord Injury Repair"

_cells, 2021, doi:10.3390/cells10040901_

Round 1
Reviewer 1 Report
The authors describe the current state of the art in the field of mesenchymal stem cells and their applications in regenerative medicine. This review covers the different types of MSC by origin, isolation and culture, cellular and immunological properties and their potential in SCI cellular therapies. The review is comprehensive and provides an excellent introduction to the primary literature. The authors note that several SCI clinical trials are underway but we are not informed about their current status. If any of these clinical studies have come to solid conclusions, it would be desirable to include a few sentences on this.
Author Response
We thank the reviewer for their comments and are happy that they felt it was a comprehensive review.
- The authors note that several SCI clinical trials are underway but we are not informed about their current status. If any of these clinical studies have come to solid conclusions, it would be desirable to include a few sentences on this.
We have included the following regarding the outcome of the clinical studies on line 71:
In general, MSC therapy after SCI has shown both an anti-inflammatory and pro-regenerative capacity in human and animal models [21, 22] and importantly their use in phase I/II clinical SCI trials have confirmed their safety [23-25]. However, whilst many clinical trials report promising efficacy for sensorimotor function, the risk of bias is high since few studies have included control groups and study subject number is low [24].
Reviewer 2 Report
The manuscript review entitled "The therapeutic potential of niche-specific mesenchymal stromal cells for spinal cord injury repair" lead by Lindsay & Barnett covers the bibliography of mesenchymal/stromal cells for graft, focusing in the use and isolation of mesenchymal stem cells and their application to spinal cord injury. The work is well written and is a significant review to settle the basis to dive in deep to the current problematic and graft failures.
MAJOR COMMENTS
The text lacks of recent bibliography and the high number of typo errors and some bibliography mistakes makes the impression to this reviewer that the manuscript has been hastily sent without checking every detail and needs a major revision.
MINOR COMMENTS:
Lines 490-493: The sentence: "OM- 490 MSCs have high mitotic activity, they maintain self-renewal ability for long periods by conserving telomeric activity and inhibiting apoptotic activity, characteristics that are not affected by donor age [143]." There is no bibliographic data in the reference number 143 (line 802).
Check some typographic errors such as:
Line 172: "5 × 106 cells" (modify the superscript of millions).
Line 282: "lower senescence ratio, [83, 84], and retain differentiation" (the use of commas)
Line 247: "VEGF-A, [69], along with BDNF" (the use of commas)
Line 248: "compared to BM-MSCs[70]" (add the space)
Line 249: "SCI repair [69]. It has been reported" (remove the extra space)
And so on, so on...
Author Response
We thank the reviewer for their comments and are happy that they felt the work is well written and a significant review.
- We apologise for the high number of typo errors and bibliography mistakes. There seems to have been an issue when our original word document was converted into the paper format. It has caused many of the issues that this reviewer refers to. Reference number 3 in the bibliography text was skipped, leaving a missing a reference at the end of the list and therefore the appearance of wrong numbers cited throughout. This was a formatting error which was not present in our word document upon uploading. In addition, we have noticed that there was variable spacing, a loss of italics, bolding and superscripting throughout the text upon conversion to the paper format. We have gone through the manuscript text and changed any errors.
- Lines 490-493: " There is no bibliographic data in the reference number 143 (line 802). This has now been resolved.
- All typographic errors, such as spacing or use of commas should now be resolved.
Line 172: "5 × 106 cells" (modify the superscript of millions).
Line 282: "lower senescence ratio, [83, 84], and retain differentiation" (the use of commas)
Line 247: "VEGF-A, [69], along with BDNF" (the use of commas)
Line 248: "compared to BM-MSCs[70]" (add the space)
Line 249: "SCI repair [69]. It has been reported" (remove the extra space)